# Dietary Patterns and Risk of Chronic Obstructive Pulmonary Disease among Chinese Adults: An 11-Year Prospective Study

**DOI:** 10.3390/nu14050996

**Published:** 2022-02-26

**Authors:** Wei Yu, Lang Pan, Weihua Cao, Jun Lv, Yu Guo, Pei Pei, Qingmei Xia, Huaidong Du, Yiping Chen, Ling Yang, Junshi Chen, Canqing Yu, Zhengming Chen, Liming Li

**Affiliations:** 1Department of Epidemiology and Biostatistics, School of Public Health, Peking University, Beijing 100191, China; weiyu@stu.pku.edu.cn (W.Y.); panlang@pku.edu.cn (L.P.); caoweihua60@163.com (W.C.); epi.lvjun@vip.163.com (J.L.); lmleeph@vip.163.com (L.L.); 2Center for Public Health and Epidemic Preparedness & Response, Peking University, Beijing 100191, China; 3Key Laboratory of Molecular Cardiovascular Sciences, Peking University, Ministry of Education, Beijing 100191, China; 4National Center for Cardiovascular Diseases, Fuwai Hospital, Chinese Academy of Medical Sciences, Beijing 100037, China; guoyu@kscdc.net; 5Chinese Academy of Medical Sciences, Beijing 100730, China; peipei@kscdc.net (P.P.); xiaqingmei@kscdc.net (Q.X.); 6Clinical Trial Service Unit and Epidemiological Studies Unit (CTSU), Nuffield Department of Population Health, University of Oxford, Oxford OX3 7LF, UK; huaidong.du@ndph.ox.ac.uk (H.D.); yiping.chen@ndph.ox.ac.uk (Y.C.); ling.yang@ndph.ox.ac.uk (L.Y.); zhengming.chen@ndph.ox.ac.uk (Z.C.); 7Medical Research Council Population Health Research Unit, University of Oxford, Oxford OX3 7LF, UK; 8China National Center for Food Safety Risk Assessment, Beijing 100022, China; chenjunshi@cfsa.net.cn

**Keywords:** dietary pattern, chronic obstructive pulmonary disease, COPD, risk of morbidity

## Abstract

The evidence about the association between dietary patterns and the incidence of chronic obstructive pulmonary disease (COPD) among Chinese adults is limited. In the present study, we analyzed the prospective data of 421,426 participants aged 30–79 years from the China Kadoorie Biobank. Factor analysis with a principal component method was employed to identify dietary patterns. Cox proportional hazard regression models were performed to explore the association between dietary patterns and incident COPD. Two dietary patterns were identified: the traditional northern dietary pattern was characterized by a low intake of rice and a high intake of wheat and other staple foods, while the balanced dietary pattern was characterized by a high intake of fresh fruit and protein-rich foods (soybean, meat, poultry, fish, eggs, and dairy products). During a median follow-up of 11.13 years, 5542 men and 5750 women developed COPD. After adjustments for potential confounders, the balanced dietary pattern was associated with a lower risk of COPD (*p* for trend <0.001), with a hazard ratio (95% confidence interval) of 0.75 (0.67, 0.84) for those in the highest quintile compared with those in the lowest quintile. Such association was modified by sex, smoking status, and adiposity level.

## 1. Introduction

Chronic obstructive pulmonary disease (COPD) is a progressive respiratory disease and is often accompanied by other medical conditions, thus resulting in a poor quality of life [1]. COPD is the third leading cause of death worldwide [2]. In China, it is estimated that nearly 100 million people suffer from COPD, accounting for 8.6% of adults aged 20 years or older [3]. Therefore, importance should be attached to the prevention of COPD.

Diet is one of the modifiable risk factors for COPD. Previous studies have focused on the independent effect of individual foods or nutrients on COPD [4]. However, it is difficult and almost impracticable to eliminate the impact of other foods consumed together. Thus, dietary pattern, a measurement of the whole diet, has been proposed [5].

Previous studies have reported the association of dietary patterns and COPD risk [6,7,8,9,10] or lung function [8,9,10,11,12]. For example, two typical dietary patterns were identified among US health professionals, the “prudent pattern” and the “western pattern”, respectively, that showed an inverse association and positive association between COPD risks [6,7]. Another research in the US observed similar results [8]. A study in the UK population reported that a similar “prudent pattern” was associated with higher lung function and lower COPD prevalence [9]. However, most earlier studies were cross-sectional studies with a limited sample size. Evidence from higher-quality studies, such as population-based prospective study and experimental study, is urgently required to verify the relationship of dietary patterns and COPD risk. Furthermore, existing research was mainly conducted in Europe and North America. Due to ecological variability and traditional culture, dietary habits differ enormously across countries and regions. Thus, it is essential to construct dietary patterns that fully capture the characteristics of regional diet.

Chinese cuisine is quite distinctive from other countries and have dramatically changed in recent decades. The present study examined the relationship between dietary patterns and COPD risk among Chinese adults in a large prospective study.

## 2. Materials and Methods

### 2.1. Study Population

The present study was based on the China Kadoorie Biobank (CKB). The detailed introductions of the CKB study have been previously reported [13,14]. Briefly, the CKB study is a large prospective cohort, primarily aiming to reveal effects of various risk factors (genetics, lifestyle, environment, etc.) for major chronic diseases. We included about 0.5 million Chinese adults from five urban (Harbin, Qingdao, Suzhou, Haikou, and Liuzhou) and five rural regions (Gansu, Sichuan, Hunan, Henan, and Zhejiang). The study regions were carefully selected according to the prevalence of common chronic diseases and risk factors, level of economic development, quality of local infrastructure, and the relative stability of the population. Permanent residents aged 30–79 without severe disability in the selected regions were eligible participants. A total of 512,726 respondents completed the baseline survey between 2004 and 2008, of which the main contents included an interviewer-administered questionnaire, physical examination, and blood draw. Since recruitment, resurveys involving about 5% of the participants have been conducted every 4 or 5 years, using a procedure broadly consistent with the baseline survey. Up until now, two resurveys have been completed, and the third resurvey is ongoing.

In the present study, we excluded participants with prevalent COPD at baseline (n = 37,057), i.e., either clinically-identified COPD or screen-detected COPD. Clinically-identified COPD was defined as self-report of physician diagnosis of chronic bronchitis or emphysema. Screen-detected COPD was defined according to Global Initiative for Chronic Obstructive Lung Disease criteria as the ratio of forced expiratory volume in 1 s (FEV1) over forced vital capacity (FVC) <0.7 (i.e., FEV1/FVC <0.7), in addition to no reports of physician diagnosis of COPD at the baseline survey. Furthermore, we also excluded participants with physician-diagnosed asthma (n = 1566), tuberculosis (n = 6383), coronary heart disease (n = 13,275), stroke (n = 6785), and cancer (n = 2132), and those with prevalent diabetes (n = 23,757) defined by self-report or onsite plasma glucose testing (fasting blood glucose ≥7.0 mmol/L or random blood glucose ≥11.1 mmol/L) at baseline, due to the fact that these medical conditions might affect dietary habits and the incidence of COPD. Moreover, respondents with an abnormal value of FEV1/FVC (>1) (n = 343) and those with a missing value of body mass index (BMI) (n = 2) were also excluded, leaving 421,426 subjects eligible for the present analysis.

Both the Ethical Review Committee of the Chinese Center for Disease Control and Prevention (Beijing, China) and the Oxford Tropical Research Ethics Committee at University of Oxford (UK) approved the CKB study, and all participants provided written informed consent.

### 2.2. Assessment of Dietary Intake

At baseline, participants were asked to indicate the intake frequency of common food groups in China during the last year, which included rice, wheat, other staple foods (such as corn, millet, etc.), meat, poultry, fish or sea food, fresh eggs, fresh vegetables, preserved vegetables, soybean, fresh fruit, and dairy products. The frequency ranged from “never or rarely”, “monthly”, “1–3 days/week”, “4–6 days/week” to “daily”, which was converted to 0, 0.5, 2, 5, and 7 days/week, respectively. At the second resurvey, we further obtained the amount of daily intake of each food group, which was used to estimate the weekly amount of food consumption and daily energy intake.

In addition, the intake frequency, type, and amount of tea (green/jasmine tea, oolong tea, black tea, or other tea) and alcohol (beer, rice wine, wine, spirit with ≥40% alcohol, or spirit with <40% alcohol) were collected, since both of them have been traditional beverages in China. Hence, the average amount of weekly consumption of tea (in g/week) and alcohol (in g/week) was estimated accordingly.

The reliability of the food and beverage questionnaire was confirmed by a repeated questionnaire survey among 926 respondents within one year after the baseline. For most of the food groups and beverages, the age- and sex-adjusted spearman correlation coefficients were >0.50. In particular, rice, wheat, and oolong tea showed high correlation with baseline measures (0.74, 0.76, and 0.71, respectively) [15].

### 2.3. Assessment of Covariates

The baseline questionnaire included socio-demographic (region, sex, age, education, marital status, and household income), health-related behavior (tobacco smoking, alcohol consumption, physical activity, and nutritional supplements), and indoor air pollution (passive smoking and household air pollution). In detail, the usual type and duration of physical activities for occupation, commuting, housework, and leisure-time exercise in the past year were recorded, then the metabolic-equivalent task hour (MET-h) was calculated accordingly. Nutritional supplements included fish oil/cod liver oil, vitamins, calcium/iron/zinc, ginseng, and other herbal products. Passive smoking was evaluated by the duration of living with a smoker (in year) and exposure to smoke (in hour/week). Household air pollution consisted of cooking pollution and heating pollution, and both were assessed by the use of solid fuel (i.e., coal and wood) for cooking and heating.

Additionally, trained staff measured standing height and weight with standard protocol and calibrated instruments. BMI was calculated as weight divided by height square (kg/m^2^).

### 2.4. Ascertainment of Incident COPD

After the baseline survey, long-term follow-up was carried out until the date of COPD (International Classification of Diseases-10 J41-J44) incidence, death, loss to follow-up, or 31 December 2017, whichever came first. Information on COPD incidence was obtained mainly via the national health insurance (HI) system with linkage to hospitalization records. For participants who failed to establish a link to the HI system (about 2%), professional investigators performed active follow-up annually to ascertain if participants were newly diagnosed with COPD last year. Vital status and cause of death were confirmed by checking against death registries and official residential records.

In order to ensure the validity of COPD diagnosis, 1069 COPD cases were randomly selected and independently reviewed by five experienced physicians. The results showed that the positive predictive value of COPD diagnosis in CKB study was 85% [16].

### 2.5. Statistical Analysis

Factor analysis with principal component analysis was performed to obtain dietary patterns from the twelve food groups (in days/week) and nine drinks (in g/week), i.e., the factors extracted by factor analysis represented dietary patterns in our study. Moreover, the factors were transformed by varimax rotation to achieve better interpretability. The number of factors was determined according to eigenvalues, scree plot, and interpretability. Factor loading indicated the intensity and direction of the association between the factor and food groups. Moreover, food groups with an absolute factor loading of ≥0.4 were considered to make a contribution to the factor, which is the foundation to the name of the dietary patterns. Additionally, the factor scores of each factor and each participant were calculated by summing the consumption of food groups weighted by factor loading. A higher score indicated higher compliance to the corresponding dietary pattern. The score was divided into quintiles (Q1 to Q5 in ascending order) in subsequent analysis.

To describe baseline characteristics, linear regression (for continuous variables) or logistic regression model (for categorical variables) were used to calculate mean or percentage across the quintile of each dietary pattern, with adjustment for age, sex, and region. Cox proportional hazard models were conducted to estimate the hazard ratio (HR) and 95% confidence interval (CI) between dietary pattern and incident COPD, with stratification on age at baseline (in 5-year interval) and ten study regions and with age as the time scale. The proportional hazard assumption was checked by the Kaplan–Meier survival curve and the significance of the interaction term between dietary pattern and time. Three multivariate models were established to adjust potential confounders. Model 1 was adjusted for sex (male or female), education level (no formal school, primary school, middle school, high school, or college/university), marital status (married or other), and household income (<10,000, 10,000–19,999, or ≥20,000 Chinese yuan (CNY)/year). Model 2 was additionally adjusted for smoking status (never/occasional, former, and having quit ≥5 years or <5 years, current and 1 to 14 cigarettes/day, 15 to 24 cigarettes/day, or ≥25 cigarettes/day), alcohol consumption (ex-regular drinkers, not weekly drinking, weekly but not daily drinking, daily and <15 g/day, 15–29 g/day, 30–59 g/day, or ≥60 g/day), physical activity (continuous, MET-h), nutritional supplements (yes or no), daily energy intake (continuous in log-transformed formal, kJ/day), and BMI (continuous, kg/m^2^). Model 3 further added passive smoking (never lived with a smoker, lived with a smoker for <20 years, lived with a smoker for ≥20 years and exposure <20 h/week, or lived with a smoker for ≥20 years and exposure ≥20 h/week), cooking fuel pollution (never or occasionally cook, daily cook with clean fuel, daily cook with solid fuel, or daily cook with other fuel), and heat fuel pollution (never or occasionally heat, heat with clean fuel, heat with solid fuel, or heat with other fuel).

We conducted several sensitivity analyses to test the robustness of the results. First, we estimated the portion size of each food group by the second resurvey according to region, sex, and age, and calculated the weekly amount of food consumption (g/week). Then, the main analysis was rerun. Second, we used the lower limit of normal (LLN) definition rather than FEV1/FVC <0.7 to exclude participants with airflow obstruction at baseline. Third, we excluded newly diagnosed COPD cases during the first 2 years of follow-up to avoid subclinical cases. Fourth, we additionally adjusted for waist circumference. Fifth, we used propensity scores estimated by multinomial logistic models to adjust for confounders. Finally, for women, we further adjusted reproductive history, including age at menarche, menopause status, the use of oral contraceptive pills, the history of pregnancy, and the history of gynecological surgery (hysterectomy, oophorectomy, or mastectomy).

Subgroup analyses were conducted to explore the modifiable effect of eight baseline variables on the relationship between dietary pattern score (in continuous) and incident COPD. The interaction between the stratifying variable and dietary pattern was determined by the significance of their cross-product term. The adjusted significance level for interaction was 6.25 × 10^−3^ by using the Bonferroni method.

All statistical analyses were conducted with SAS version 9.4 (SAS Institute Inc., Cary, NC, USA). Statistical significance was set as *p* < 0.05, except for interaction tests.

## 3. Results

Two dietary patterns were identified, and they explained 24.45% of the variance of twelve food groups and nine beverages (Table 1). The first pattern was characterized by a high intake of wheat and other staple foods (such as corn, millet, etc.) and a low intake of rice, and it was named as the traditional northern dietary pattern. The second pattern was characterized by high consumption of fresh fruit, soybean, meat, poultry, fish or sea food, eggs, and dairy products, and it was called the balanced dietary pattern. The detailed characteristics of dietary patterns by quintile categories are shown in Appendix A. Similar dietary patterns were derived using the estimated food consumption amount via the second resurvey. The weighted kappas for two dietary patterns were 0.79 and 0.76, respectively (Appendix A).

Among 421,426 individuals, the mean age was 50.64 ± 10.28, 59.85% were women, 42.54% lived in the urban area, and 60.06% inhabited in the south. As shown in Table 2, participants with higher scores of the traditional northern dietary pattern tended to reside in the north, to have a higher level of education, and to use nutritional supplements, but they were less likely to smoke weekly and less likely to be exposed to secondhand smoke and cooking fuel pollution. Compared to respondents with lower scores of the balanced dietary pattern, those with higher scores were more likely to be younger, from an urban area, married, to receive a higher education level, to have a higher annual household income, to drink weekly, and to use nutritional supplements, but they were less likely to use solid fuel for cooking.

During a median follow-up of 11.13 years (about 4.58 million person-years), 11,292 newly diagnosed COPD cases were documented. The crude incidence rate was 2.47 per 1000 person-years. After adjustment for potential confounders, no statistically significant associations were found between the traditional northern dietary pattern and incident COPD among total participants, nor were any found among men and women (*p* for sex interaction = 0.159; all *p* for trend >0.05) (Table 3).

As for the balanced dietary pattern, it was negatively associated with the risk of COPD among total population (*p* for trend <0.001). As compared with the lowest quintile (Q1), HRs (95% CIs) for Q2 to Q5 were 0.89 (0.84, 0.94), 0.87 (0.81, 0.93), 0.84 (0.77, 0.91), and 0.75 (0.67, 0.84), respectively. Such association was differed by sex (*p* for interaction <0.001), although HRs between men and women were similar (*p* for trend: men 0.007, women 0.031). The risk of newly diagnosed COPD for the highest quintile decreased 19% (HR: 0.81; 95% CI: 0.69, 0.96) and 21% (HR: 0.79; 95% CI: 0.66, 0.94) for men and women, respectively, as compared to the lowest quintile of the balanced dietary pattern (Table 3).

In sensitivity analyses, the results did not change substantially when we reconstructed dietary patterns with the estimated food consumption amount; excluded participants with airflow obstruction at baseline defined by LLN criteria; excluded incident COPD cases within the first 2 years of follow-up; further adjusted for waist circumference; used propensity scores to adjust for confounders; further adjusted for reproductive history in women (Appendix A).

The HRs (95% CIs) per standard deviation (SD) increment of dietary pattern score for each subgroup are displayed in Figure 1. The association between the traditional northern dietary pattern and COPD risk differed in regions (rural or urban) (*p* for interaction <0.001). For the balanced dietary pattern, we found that the association was stronger among nonweekly smokers (HR: 0.86; 95% CI: 0.81, 0.92) than weekly smokers (HR: 0.92; 95% CI: 0.86, 0.98), and similar results were observed in the analysis by BMI categories (*p* for interaction <0.001 and 0.006, respectively); HRs (95% CIs) per SD for underweight, normal, overweight, and obesity participants were 0.97 (0.84, 1.12), 0.84 (0.79, 0.90), 0.91 (0.83, 0.99), and 0.97 (0.84, 1.13), respectively.

## 4. Discussion

In the large prospective cohort with 11 years of follow-up among Chinese adults, a balanced dietary pattern was identified; it was rich in fresh fruit, soybean, meat, poultry, fish or sea food, eggs, and dairy products. The balanced dietary pattern showed a protective effect on COPD risk, with 25% lower risk for the highest intake group compared with the lowest. Such an association was modified by smoking status and adiposity level. No significant association was observed between the traditional northern dietary pattern (a high intake of wheat and other staple foods, and a low intake of rice) and risk of COPD.

In the present study, the two dietary patterns were constructed based on common food groups and beverages in China, and they fully captured the features of Chinese diet. Their characteristics are similar to previous nationwide studies, such as the Chinese Health and Nutrition Survey [17,18]. The Chinese diet varied remarkably due to ecological variability, traditional culture, and recent transition during urbanization. The traditional southern diet mainly selected rice as a staple, while the northern diet chose wheat and other staple foods. On the other hand, the balanced dietary pattern, characterized by a high intake of fresh fruit and protein-rich food (soybean, meat, poultry, fish or sea food, eggs, and dairy products), emerged along with the improvement of economy and health awareness. Both dietary patterns were in line with nutrition transition over decades in China [19].

We found that participants who had a higher balanced dietary pattern score, i.e., ate more fresh fruit and protein-rich food (soybean, meat, poultry, fish or sea food, eggs, and dairy products), had a lower risk of COPD. Existing evidence supported our findings. A recent meta-analysis included four studies and found a nonlinear negative relationship between fruit intake and COPD risk [20]. Compared with 0.1 servings/day of fruit intake, the RRs (95% CIs) for 1 and 2 servings per day were 0.78 (0.67, 0.92) and 0.70 (0.59, 0.82), respectively, although the RR of COPD decreased slightly for more than 2 servings per day [20]. The beneficial effect of fresh fruit may be partly attributed to the antioxidants and anti-inflammatory components, such as vitamin C, β-carotene, flavonoids, and dietary fiber [4]. Additionally, protein, another abundant nutrient of the balanced dietary pattern, has been associated with better lung function [21,22]. Dietary protein may alleviate COPD by maintaining the mass and strength of muscles, especially respiratory muscles. In addition, isoflavones in soy products [23] and n-3 polyunsaturated fatty acids in fish [24] may also contribute to a reduction in COPD risk. Further studies are needed to confirm our results.

We did not observe a significant relationship between the traditional northern dietary pattern (a high intake of wheat and other staple foods and a low intake of rice) and the newly developed COPD. Although several cross-sectional studies reported a beneficial effect of whole grain and harmful influence of rice on lung function, limitations of reverse causality and residual confounding should not be ignored [21,25]. Moreover, our results were consistent with a case–control study among male smokers [26]. Other foods consumed with staple foods may partly explain the difference in findings. More evidence is required to explain our results.

Of note, we observed the association between the balanced dietary pattern and COPD risk, which was significantly weakened in weekly smokers and participants with abnormal weight in subgroup analyses. It is known that tobacco smoking is the leading risk factor for COPD. Our results suggested that smoking cessation was still of great significance while keeping a healthy dietary habit. Existing literature showed that underweight participants had a higher risk of COPD; possible explanations may include lower muscle strength and more susceptibility to infection [27,28]. Several studies revealed that obesity, in particular, abdominal adiposity, was associated with higher COPD risk [27,28]. Adiposity may be accompanied by more pro-inflammatory cytokines, thereby accelerating the development of COPD. The present study suggested that a balanced dietary pattern may help prevent COPD, but the protective effect of diet may be partly weakened by abnormal BMI. Hence, developing a healthy dietary habit and maintaining a normal weight should be taken into account together in the prevention of COPD.

To the best of our knowledge, this is the first and largest prospective cohort study to examine the relationship between dietary patterns and the risk of COPD in a Chinese population. The quality of the present study was assured by strict exclusion criteria, careful adjustment for potential confounders, and multiple sensitivity analyses. A few limitations should be elucidated. First, the food frequency questionnaire was self-designed to cover twelve common food groups in China but did not collect information on the subsiding foods of each food group. However, concise questions could improve the accuracy of response and reduce measurement bias. The reliability of the food questionnaire was confirmed by a repeated survey among 926 respondents within one year after the baseline [15]. Second, we only collected the frequency but not the amount of food intake in the baseline survey. However, similar dietary patterns were derived using the estimated food consumption amount via the second resurvey. Third, when performing factor analysis, there were some arbitrary choices with respect to constructing dietary patterns, including the number of factors, the value of meaningful factor loading, and the name of dietary patterns. However, the two dietary patterns we constructed could fully capture the characteristics of the Chinese diet and were similar to previous studies [17,18]. Fourth, there might be a misclassification bias for incident COPD. For one thing, only 2.6% of Chinese adults with spirometry-defined COPD had seen a doctor [3], suggesting that it was hardly avoidable to underestimate the incidence of COPD despite diverse approaches to follow-up in our study. For another, the utility of spirometry was low among newly diagnosed COPD cases in the CKB study (13.9%). Similarly, the China Pulmonary Health study pointed out that only 12% of patients with COPD reported a previous lung function test, and the percentage was lower among COPD patients aged 40 years or older (5.9%), indicating that this is a common problem in China [3,29]. In addition, lacking spirometry evidence of airflow limitation of the incident cases restricted the possibility of exploring the dose relationship between diet and severity of airflow limitation in the present study. Nevertheless, most COPD cases (85%) in the CKB study was confirmed by comprehensive evidence, including the history of exposure to risk factors (tobacco smoking, occupational exposure, and household air pollution), respiratory symptoms, and radiological examinations [16]. Finally, although a great number of confounders were adjusted in our analysis, residual confounding factors, such as hormone replacement therapy, could not be excluded.

## 5. Conclusions

A balanced dietary pattern that is rich in fresh fruit, soybeans, meat, poultry, fish or sea food, eggs, and dairy products may lower the risk of COPD in Chinese adults. Tobacco smoking and abnormal weight partly weaken the effect; hence, quitting smoking and maintaining a normal weight while developing a healthy dietary habit are recommended actions. 

## Figures and Tables

**Figure 1 nutrients-14-00996-f001:**
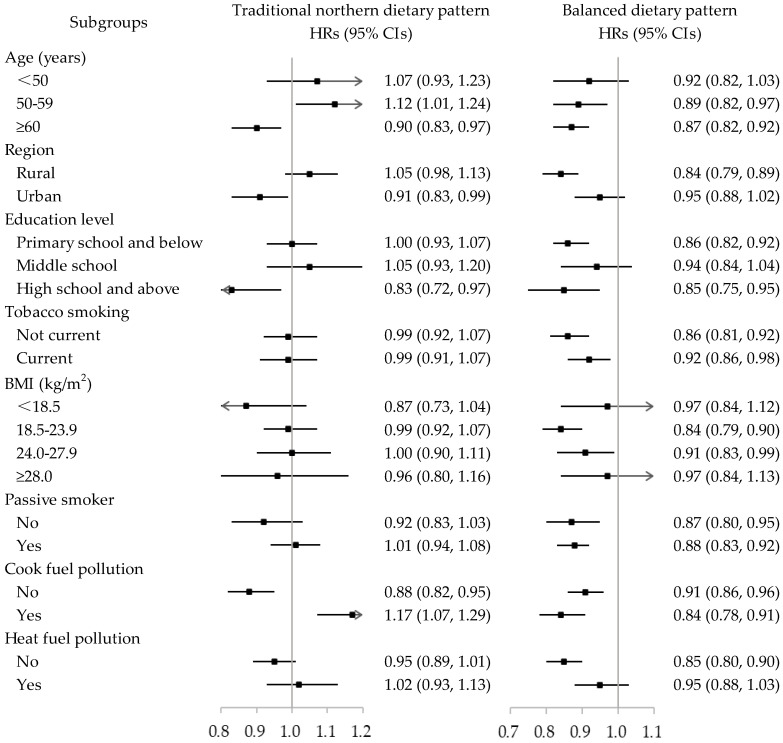
Subgroup analyses of the association between two dietary patterns and risk of COPD. Note: HRs (95% CIs) per standard deviation change of each dietary pattern score for Model 3 were presented. Adjusted variables included sex, education level, marital status, household income, tobacco smoking, alcohol consumption, nutritional supplement, BMI, physical activity, daily energy intake, passive smoking, cook fuel pollution, and heat fuel pollution, if appropriate. *p* values for interaction were <6.25 × 10^−3^ for region (*p* < 0.001) in the traditional northern dietary pattern and tobacco smoking (*p* < 0.001) and BMI (*p* = 0.006) in the balanced dietary pattern. HR: hazard ratio; CI: confidence interval; CNY: unit of Chinese money Yuan; BMI: body mass index.

**Table 1 nutrients-14-00996-t001:** Factor loading matrix of major dietary patterns by principal component analysis with varimax rotation (n = 421,426).

Food or Beverage Group	Traditional Northern Dietary Pattern	Balanced Dietary Pattern
Rice	**−0.84**	0.20
Wheat	**0.84**	0.12
Other staple foods	**0.70**	−0.16
Meat	−0.38	**0.59**
Poultry	−0.36	**0.55**
Fish/sea food	−0.37	**0.51**
Eggs	0.32	**0.52**
Fresh vegetables	−0.07	0.20
Soybean	−0.14	**0.47**
Preserved vegetables	−0.14	0.13
Fresh fruit	0.02	**0.70**
Dairy products	0.22	**0.64**
Beer	0.06	0.20
Rice wine	−0.14	−0.01
Wine	<0.01	0.06
Heavy spirit (≥40%)	−0.07	<0.01
Light spirit (<40%)	−0.12	−0.02
Green tea	<0.01	0.23
Oolong tea	−0.08	0.06
Black tea	−0.19	−0.05
Other tea	−0.01	0.01
Variance explained (%)	14.83	9.62

Note: Figures in bold indicated absolute factor loading ≥0.40.

**Table 2 nutrients-14-00996-t002:** Baseline characteristic of participants according to quintiles of dietary patterns (n = 421,426).

Baseline Characteristic	Traditional Northern Dietary Pattern			Balanced Dietary Pattern		
	Q1	Q3	Q5	Q1	Q3	Q5
Age (year, $\bar{x}$)	49.11	49.41	53.46	53.08	50.44	48.92
Women (%)	48.46	58.81	65.77	69.31	57.19	56.03
Urban area (%)	50.13	49.24	17.00	6.75	37.97	86.11
Southern area (%)	98.92	85.10	0.25	43.10	73.52	47.32
Married (%)	91.63	91.37	92.19	87.45	91.67	94.12
High school and above (%)	12.71	21.61	28.85	9.72	17.04	35.36
Household income≥ 20,000 CNY/year (%)	38.54	44.13	48.55	24.27	39.62	60.29
Current smoker (%)	32.69	28.66	25.33	29.37	29.33	26.45
Current drinker (%)	22.30	13.50	11.41	11.79	15.01	17.71
Using nutritional supplement (%)	10.23	17.38	28.10	7.61	13.35	29.12
Passive smoker (%)	79.19	75.66	71.87	75.97	76.27	73.38
Cooking with solid fuel (%) ^a^	43.01	37.32	30.24	42.83	36.77	23.47
Heating with solid fuel (%) ^a^	39.54	37.94	36.79	40.43	39.37	32.10
Physical activity (MET-h/d, $\bar{x}$)	22.91	21.89	21.75	22.88	22.47	20.93
BMI (kg/m^2^, $\bar{x}$)	23.69	23.69	23.40	23.30	23.63	23.79

Note: The results were shown as adjusted means or percentages, with adjustment for age, sex, and region, if appropriate. CNY: unit of Chinese money Yuan; MET: metabolic equivalent task; BMI: body mass index. ^a^ Solid fuel included coal and wood.

**Table 3 nutrients-14-00996-t003:** HRs (95% CIs) for the association between dietary patterns and risk of COPD.

	Quintile of Dietary Pattern Scores					*p* for Trend
	Q1	Q2	Q3	Q4	Q5	
Traditional northern dietary pattern						
Total (n = 421,426)						
Case	2292	3419	3142	1594	845	
Cases/1000 PYs	2.48	3.75	3.47	1.75	0.91	
Model 1	1.00	1.07 (1.01, 1.13)	1.06 (1.00, 1.12)	0.99 (0.90, 1.08)	0.89 (0.78, 1.02)	0.199
Model 2	1.00	1.06 (1.00, 1.12)	1.06 (1.00, 1.13)	1.01 (0.92, 1.12)	0.91 (0.79, 1.04)	0.469
Model 3	1.00	1.06 (1.00, 1.12)	1.06 (1.00, 1.13)	1.02 (0.92, 1.12)	0.91 (0.79, 1.05)	0.507
Men (n = 169,188)						
Case	1278	1495	1549	767	453	
Cases/1000 PYs	3.02	4.81	4.47	2.17	1.21	
Model 1	1.00	1.04 (0.96, 1.13)	1.02 (0.94, 1.11)	0.93 (0.82, 1.05)	0.88 (0.73, 1.06)	0.136
Model 2	1.00	1.05 (0.97, 1.14)	1.01 (0.92, 1.10)	0.90 (0.79, 1.04)	0.84 (0.69, 1.02)	0.060
Model 3	1.00	1.05 (0.97, 1.14)	1.01 (0.92, 1.10)	0.91 (0.79, 1.04)	0.84 (0.69, 1.03)	0.065
Women (n = 252,238)						
Case	1014	1924	1593	827	392	
Cases/1000 PYs	2.02	3.21	2.85	1.48	0.71	
Model 1	1.00	1.07 (0.99, 1.16)	1.09 (1.00, 1.19)	1.07 (0.94, 1.22)	0.91 (0.75, 1.11)	0.986
Model 2	1.00	1.07 (0.99, 1.16)	1.13 (1.04, 1.23)	1.19 (1.04, 1.37)	1.05 (0.85, 1.28)	0.097
Model 3	1.00	1.07 (0.99, 1.16)	1.13 (1.03, 1.23)	1.19 (1.03, 1.37)	1.04 (0.85, 1.28)	0.104
Balanced dietary pattern						
Total (n = 421,426)						
Case	3062	2789	2504	1818	1119	
Cases/1000 PYs	3.38	3.05	2.74	1.98	1.21	
Model 1	1.00	0.90 (0.86, 0.95)	0.90 (0.85, 0.95)	0.88 (0.82, 0.94)	0.80 (0.74, 0.87)	<0.001
Model 2	1.00	0.89 (0.84, 0.94)	0.86 (0.81, 0.92)	0.83 (0.76, 0.90)	0.74 (0.66, 0.83)	<0.001
Model 3	1.00	0.89 (0.84, 0.94)	0.87 (0.81, 0.93)	0.84 (0.77, 0.91)	0.75 (0.67, 0.84)	<0.001
Men (n = 169,188)						
Case	1160	1381	1366	998	637	
Cases/1000 PYs	3.90	3.94	3.56	2.49	1.69	
Model 1	1.00	0.97 (0.90, 1.06)	0.96 (0.89, 1.05)	0.91 (0.83, 1.00)	0.90 (0.80, 1.01)	0.032
Model 2	1.00	0.94 (0.87, 1.03)	0.91 (0.83, 1.00)	0.84 (0.75, 0.95)	0.81 (0.69, 0.95)	0.007
Model 3	1.00	0.94 (0.87, 1.03)	0.91 (0.83, 1.00)	0.85 (0.75, 0.95)	0.81 (0.69, 0.96)	0.007
Women (n = 252,238)						
Case	1902	1408	1138	820	482	
Cases/1000 PYs	3.13	2.49	2.14	1.58	0.88	
Model 1	1.00	0.90 (0.83, 0.96)	0.89 (0.82, 0.96)	0.90 (0.82, 0.99)	0.76 (0.67, 0.86)	<0.001
Model 2	1.00	0.90 (0.83, 0.97)	0.90 (0.82, 0.99)	0.91 (0.81, 1.03)	0.78 (0.65, 0.92)	0.018
Model 3	1.00	0.90 (0.83, 0.98)	0.90 (0.82, 0.99)	0.92 (0.82, 1.04)	0.79 (0.66, 0.94)	0.031

Note: Adjusted HRs (95% CIs) are presented. Model 1 was adjusted for sex (only in total population), education level, marital status, and household income. Model 2 was further adjusted for tobacco smoking, alcohol consumption, nutritional supplement, BMI, physical activity, and daily energy intake. Model 3 additionally included passive smoking, cook fuel pollution, and heat fuel pollution. *p* values for interaction between sex and each dietary pattern were 0.159 and <0.001 for the traditional dietary northern pattern and the balanced dietary pattern, respectively. HR = hazard ratio; CI = confidence interval; PY = person-year; BMI = body mass index.

## Data Availability

The access policy and procedures are available at www.ckbiobank.org (accessed on 15 January 2022).

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
