# Peer review of "Dietary Patterns and Risk of Chronic Obstructive Pulmonary Disease among Chinese Adults: An 11-Year Prospective Study"

_nutrients, 2022, doi:10.3390/nu14050996_

Round 1

Reviewer 1 Report

This is a well written paper that has used appropriate statistical  methods to show a relationship between a balanced diet and development  of COPD.

The two changes I would like to see are a propensity score analysis to further adjust for confounders. Second, the recognition that self reported COPD is not very reliable and that misclassification based on this approach is a limitation of the study.  A further limitation of not having spirometric evidence of airflow limitation to confirm incident COPD is that any dose relationship between diet and severity of airflow limitation cannot be explored in this study.

These issues should be discussed..

Reviewer 2 Report

Thank you for the opportunity to review in a good journal. I am very sorry for the late review due to the corona-19 infection.

This is an interesting study that looked at the relationship between COPD and food.

It is a study that I have been interested in for a long time. However, there are serious questions involving study design.

  1. Many factors must be adjusted to determine the association between diet and COPD incidence. (Region, society, comorbidities, female history, hormones, medications taken, fine dust, etc.) Some of these are missing from the current adjustment variable.

For example, it is known that female hormones may play a protective role in the development of lung disease. In this study, menopause was included in the adjusted factor, but drugs such as hormone therapy and oral contraceptive were not included in the adjusted factor.

  1. Previous guidelines were for adults over 40 years of age for COPD, and now they are over 20 years old. Is there a reason why you decided to be 30-70 years old?
  2. “In the present study, participants with prevalent COPD were excluded (n = 37,057)” -> What is the mean? (prevalent COPD?)
  3. This report show to exclude comorbidities that may be associated with COPD. (TB, asthma, DM, etc.) Are bronchiectasis and lung cancer included? If yes, why include?
  4. The first COPD in the abstract should be written in full term, not an abbreviation.
  5. "hazard ratio 0.75 (0.67, 0.84) for those in the highest quintile" The range in parentheses is shown as CI, and should be written as the full term when it first appears.

Round 2

Reviewer 2 Report

Thank you for the revised manuscript. I have no further comments.